# Insulin-Degrading Enzyme Regulates mRNA Processing and May Interact with the CCR4-NOT Complex

**DOI:** 10.3390/cells14110792

**Published:** 2025-05-28

**Authors:** Barbara Bertocci, Ayse Yilmaz, Emmanuelle Waeckel-Énée, Chiara Guerrera, Kevin Roger, Lamine Touré, Peter M. van Endert

**Affiliations:** 1Université Paris Cité, INSERM, CNRS, Institut Necker Enfants Malades, F-75015 Paris, France; barbara.bertocci@inserm.fr (B.B.); ayse.yilmaz173@hotmail.fr (A.Y.); emmanuelle.enee@inserm.fr (E.W.-É.); 2Université Paris Cité, INSERM, CNRS, Structure Fédérative de Recherche Necker, Proteomics Platform, F-75015 Paris, France; chiara.guerrera@inserm.fr (C.G.); kevin.roger@inserm.fr (K.R.); 3Université Paris Cité, INSERM, CNRS, Institut Necker Enfants Malades, Bioinformatics Hub, F-75015 Paris, France; lamine.toure@inserm.fr; 4Service Immunologie Biologique, AP-HP, Hôpital Universitaire Necker-Enfants Malades, F-75015 Paris, France

**Keywords:** insulinase, protein homeostasis, RNA processing, CCR4-NOT, beta cell, islet of Langerhans

## Abstract

Insulin-degrading enzyme is a zinc metalloprotease that degrades low-molecular-weight substrates, including insulin. Ubiquitous expression, high evolutionary conservation, upregulation of Ide in stress situations, and literature findings suggest a broader function of Ide in cell physiology and protein homeostasis that remains to be elucidated. We used proteomics and transcriptomics approaches to search for leads related to a broader role of Ide in protein homeostasis. We combined an analysis of the proteome and single-cell transcriptome of *Ide^+/+^* and *Ide^−/−^* pancreatic islet cells with an examination of the interactome of human cytosolic Ide using proximity biotinylation. We observe an upregulation of pathways related to RNA processing, translation and splicing in *Ide^+/+^* relative to *Ide^−/−^* islet cells. Corroborating these results and providing a potential mechanistic explanation, proximity biotinylation reveals interaction of Ide with several subunits of CCR4-NOT, a key mRNA deadenylase regulating gene expression “from birth to death”. We propose a speculative model in which human and murine Ide cooperate with CCR4-NOT to control protein expression in proteotoxic and metabolic stress situations through cooperation between their deadenylase and protease functions.

## 1. Introduction

Insulin-degrading enzyme (Ide; also known as insulysin) is a zinc metalloprotease of 110 kDa originally described for its ability to degrade insulin [1]. Although Ide is thought to participate in the clearance of extra- and intracellular insulin [2,3], *Ide^−/−^* mice display only mild or no hyperinsulinemia, suggesting a complex role of Ide in insulin clearance [4]. Ide is mainly located in the cytosol but is also found in endosomes, mitochondria and peroxisomes [5]. Ide degrades low-molecular-weight substrates including hormones, neuropeptides and growth factors, but also amyloid β and ubiquitin [6]. Interestingly, rather than displaying sequence-dependent cleavage specificity, Ide seems to prefer substrates able to form amyloids. Remarkably, Ide displays high evolutionary conservation from yeast to plants to all eukaryotes, including organisms lacking identified cognate substrates, which, together with its ubiquitous expression in vertebrate organisms, suggests functions in cellular homeostasis that remain poorly understood [7].

Several lines of evidence reported in the literature and obtained in our laboratory suggest that Ide may play a role in cellular protein homeostasis and stress responses. Ide has been suggested to act as a “dead-end chaperone” that forms stable complexes with some substrates, including a precursor protein of varicella-zoster virus, amyloid β and α-synuclein, potentially preventing toxic aggregate formation of the latter [8,9,10]. Moreover, Ide not only binds and degrades ubiquitin [11], but also has been shown to interact with the proteasome modulating its activity [12,13], which is possibly regulated through insulin binding to Ide [14].

We have reported evidence that Ide plays a role in stress responses. Thus, *S. pombe* yeast cells lacking an Ide homologue (Iph) display an increased resistance to proteotoxic stress that is conferred through an mTOR-dependent pathway [15]. Induction of proteotoxic stress in pancreatic beta cells through proteasome inhibition or tunicamycin induces upregulation of *Ide* mRNA levels, which is abolished by mTOR inhibition [4]. Conversely, mice with genetic *Ide* deficiency display a low-level unfolded protein response (UPR) in pancreatic islet cells that is enhanced by tunicamycin [4]. Similarly, Ide inhibitors enhance the UPR triggered in hepatocytes by tunicamycin [16]. At the same time, Ide deficiency induces mTOR-dependent proliferation in pancreatic islet cells in the steady state, possibly through a pathway related to the rescue of *Iph^−/−^* yeast cells from tunicamycin stress. In C57BL/6 mice fed a high-fat diet, proliferation is enhanced with dramatic weight gain, insulin and proinsulin over-production, and development of diabetes [4]. Thus, Ide deficiency triggers a low-level stress response that is enhanced upon exposure to massive proteotoxic stress, which could be related to a role of Ide in protein breakdown and turnover. At the same time, rather than attenuating protein translation as expected, stressed *Ide^−/−^* cells activate mTOR and cell proliferation, suggesting that Ide may physiologically act to limit protein translation under conditions of metabolic or proteotoxic stress.

How could Ide limit protein translation under conditions of metabolic stress? Although, as mentioned, Ide deficiency has a very limited impact on blood insulin levels, it is possible that the enzyme limits blood insulin levels in stress situations, e.g., by preventing excessive mTOR activation in adipose tissue. Ide could also regulate protein translation through upregulation of proteasome activity, although the available literature indicates that Ide tends to inhibit rather than activate the proteasome [17]. It is also conceivable that Ide regulates protein homeostasis through protein–protein interactions independently of its proteolytic function. Consistent with such a scenario, in the *S. pombe* system mentioned above, reconstitution of *Iph^−/−^* cells with protease-dead human *IDE* reconstituted normal sensitivity to proteotoxic stress [15].

In this study, we used unbiased proteomics and transcriptomics approaches to identify potential leads related to the role of Ide in cellular protein homeostasis. In addition to re-examining our prior proteomics analysis of proteins expressed in *Ide^+/+^* and *Ide^−/−^* pancreatic islets of auto-immune non-obese diabetic (NOD) mice [4], we performed a proximity biotinylation analysis of proteins potentially interacting with Ide in the cytosol of HEK cells and examined single-cell transcriptomes of *Ide^+/+^* and *Ide^−/−^* pancreatic islet cells from non-autoimmune C57BL/6 mice. All three independent approaches provided evidence suggesting that Ide plays a role in mRNA processing, splicing, and translation. Intriguingly, we discovered that human Ide likely interacts with several subunits of the CCR4-NOT complex (including the deadenylase CNOT8), a key cytosolic and nuclear complex regulating gene expression in all eukaryotes from the production of mRNAs in the nucleus to their degradation in the cytoplasm. The large, evolutionarily conserved CCR4-NOT complex controls mRNA deadenylation, a major step in mRNA decay and turnover, and influences regulation of translation and transcription independently of its deadenylation activity. The subunits of CCR4-NOT, each of which has distinct functions, include CNOT1, a large scaffold protein of >2300 amino acids; the heterodimer formed by the structurally similar CNOT2/CNOT3 proteins that modulates the activity of the deadenylases and transcription; the paralog deadenylases CNOT6/CNOT6L and CNOT7/CNOT8, which are mutually exclusively incorporated into the complex; and additional subunits including CNOT4 (an E3 ligase), CNOT9, CNOT10, and CNOT11. CCR4-NOT deadenylases mediate targeted decay of mRNA after its initiation by the Pan2/Pan3 complex and also mediate generic (upon translation termination) and nonsense-mediated mRNA decay. mRNA decay by CCR5-NOT occurs both in P bodies and independent of them. Beyond mRNA decay, CCR4-NOT is implicated in regulation of protein levels, transcription, repair of replication stress, nuclear RNA export, the cell cycle, and senescence [18,19]. Our findings reveal an unanticipated role of Ide that may underlie its role in protein homeostasis and explain its strong evolutionary conservation.

## 2. Materials and Methods

### 2.1. Proximity Biotinylation Analysis in HEK293 Cells

#### 2.1.1. Generation of IDE-TurboID Lentiviruses and HEK Transduction

The construct encoding the cytoplasmic form of human IDE fused with V5tagTurboID was generated by replacing an insert encoding mitochondrial *Ide* in the published lentiviral construct pLVX-IDE-TurboID V5tag-P2A-T2A-tdTomato [20]. Ide cDNA, starting at the second initiation codon (Met^42^), was amplified with the addition of an XhoI at the 5′ end and an EcoRI restriction site at the 3′ end and inserted into the lentiviral construct as an XhoI-EcoRI fragment. A control construct expressing TurboID was produced by cloning a DNA fragment containing the V5tag and TurboID into the pLVX-P2A-T2A-tdTomato retroviral vector. The DNA fragment was amplified by PCR from the construct described above, adding an XhoI at the 5′ end and a MluI restriction site at the 3′ end. The primers for PCR amplification are listed in Appendix A.

Plasmid DNA was extracted and purified with the Nucleobond Xtra Midi EF kit (Macherey-Nagel) and used to produce lentiviruses with an average titer of 10^9^ TU/mL. HEK (human embryonic kidney) 293 cells were obtained from ATCC (CRL-1573). HEK cells were transduced at a multiplicity of infection of five for 6 h. Cells were selected starting 48 h after infection with 10 μg/mL puromycin. After two to three passages in selection media, the percentage of transduced cells was determined by quantifying td-Tomato-positive cells by flow cytometry.

#### 2.1.2. Verification of TurboID Constructs and Proximity Biotinylation Analysis

##### Confocal Microscopy

A total of 4 × 10^4^ cells were seeded onto glass coverslips that had been precoated with poly-D-lysine overnight. The cells were fixed for 12 min with a solution containing 4% paraformaldehyde, 0.2% glutaraldehyde, 60 mM PIPES (piperazine-N,N′-bis(2-ethanesulfonic acid)), 25 mM HEPES((4-(2-hydroxyethyl)-1-piperazineethanesulfonic acid), 10 mM EGTA (ethylene glycol-bis(β-aminoethyl ether)-N,N,N′,N′-tetraacetic acid), and 2 mM magnesium acetate and then permeabilized with 0.2%Triton X-100 in DPBS (Dulbecco’s phosphate-buffered saline) for 10 min. After blocking with 5% donkey serum in PBS-0.05% Tween-20 for 2 h at RT (room temperature), the cells were incubated sequentially with primary antibodies overnight at 4 °C (IDE, V5, TOM20, see Appendix A; all antibodies used at 1:100) and then incubated with the appropriate secondary antibodies (1:200) for 2 h at room temperature. Nuclear counterstaining was carried out using 1 μg/mL DAPI (4′,6-diamidino-2-phenylindole). The slides were mounted with Vectashield Plus Antifade media. Image acquisition was performed with a 63× oil immersion objective (NA 1.4) and a laser scanning confocal microscope (TCS SP8-3X STED; Leica Microsystems, Wetzlar, Germany). The images were processed with Icy (version 2.4.3.0).

##### Western Blot

SDS-PAGE analysis was performed as described previously [20]. Primary and secondary antibodies are listed in the Appendix A.

##### TurboID Enzymatic Protein Labeling, Extraction and Analysis of Labelled Proteins

To identify enriched biotinylated proteins, batches of 20 × 10^6^ HEK293 cells were processed according to the protocol described previously [20]. The procedures for identification, quantification of proteins by NanoLC-MS/M, and data processing after LC-MS/MS acquisition were also performed as previously published [20]. The mass spectrometry proteomics data have been deposited to the ProteomeXchange Consortium via the PRIDE [21] partner repository with the dataset identifier PXD060228.

#### 2.1.3. In Situ Proximity Ligation Assay

HEK cells were fixed and permeabilized as described for confocal microscopy. Proximity ligation assays were performed with Duolink™ reagents (Appendix A) according to the manufacturer’s instructions (Sigma-Aldrich, Saint-Quentin-Fallavier, France). Primary antibodies against CNOT2, CNOT3, CNOT8, and IDE (all diluted 1:100) (Appendix A) were incubated at 4 °C overnight in a humidity chamber. The Duolink Probes, anti-mouse plus and anti-rabbit minus, were incubated 1h at 37 °C in a pre-warmed humidity chamber. Nuclear counterstaining was carried out using DAPI (1 μg/mL). Slides were mounted using Vectashield mounting medium (Vector Laboratories). The samples were imaged through a Plan Apochromat 63/1.4 numeric aperture oil-immersion lens on a confocal microscope TCS SP8 SMD (Leica, Nanterre, France), and Z-stacks were acquired using LAS X software (Leica; version 3.5.7.23225). High-resolution images were obtained with the Icy software (version 2.4.3.0). The mean number of signals per cell was obtained by dividing the total number of PLA signals by the number of nuclei (counted manually).

### 2.2. Animals

*Ide^+/+^* and *Ide^−/−^* C57BL/6J and NOD mice were obtained as described [4]. The mice were bred and housed in an ambient-temperature room with 12/12 h light/dark cycles under specific-pathogen-free conditions. Ethical approval for animal experimentation in this project was accorded by the Ministère de l’Enseignement, de la Recherche et de l’Innovation on 8 October 2020 for a period of 5 years under the number APAFIS#27441-2019080809515121.

### 2.3. Single Cell RNAseq Analysis of Pancreatic Islet Cells

#### 2.3.1. Pancreatic Islet Cells and Single-Cell Library Preparation

Islets were isolated from the pancreas of 3 *Ide^−/−^* and 3 *Ide^+/+^* C57BL/6 mice. Ice-cold collagenase P (Roche; 0.76 mg/mL in HBSS) was injected through a catheter introduced into the pancreatic duct (2 mL/mouse). The pancreas, following collagenase instillation, was excised and digested for 10 min at 37 °C, and digestion was stopped by the addition of cold HBSS containing 10% heat-inactivated FCS. The islets were handpicked under the microscope and dissociated into single-cell suspensions by incubation in 100 μL of Accutase (Thermo Fisher, Illkirch, France) at 37 °C and 5% CO_2_ for 15 min. The reaction was blocked with 1mL HBSS/10% FCS, and the cells were filtered through a 40 μm cell strainer (Corning) and centrifuged at 300× *g* for 5 min. The cells were resuspended in 100 μL of cold Sample Buffer (BD Rhapsody) and stained with the viability markers Calcein AM (Thermo Fisher Scientific) and Draq7 (BD Bioscience, Le Pont de Claix, France) according to the BD Rhapsody Single Cell Analysis protocol. The cell concentration and viability were determined with a BD Rhapsody Scanner. Cell barcoding was performed using the Custom BD Single Cell Sample Multiplexing Kit (No. 626545, Becton Dickinson, Le Pont de Claix, France) after the same number of cells was pooled for each genotype.

#### 2.3.2. Library Preparation and Data Acquisition

Cell capture and cDNA library amplification were carried out using the BD Rhapsody Cartridge (No. 66487) and Enhanced Cartridge Reagent Kit (No. 633733) and the BD Rhapsody cDNA Kit (No. 633773), respectively, according to the manufacturer’s instructions. The DNA library, from 14,755 single cells captured in a single run with four pooled barcoded samples, was constructed using the BD Rhapsody Whole Transcriptome Analysis (WTA) Pipeline version 2 (No. 633801). The single-cell library was quantified by Qubit 4 Fluorometer (Thermo Fisher) and by capillary electrophoresis using Fragment Analyzer (Agilent Technologies, Les Ulis, France). The sequencing was performed on a NovaSeq 6000 system (Illumina, Paris, France) with 3% control PhiX library (Illumina) on a Flow Cell NovaSeq S1 100cycles to generate ~1500 million paired-end reads (51 bases + 71 bases). The generated FASTQ files were uploaded to the SevenBridges platform using the command-line interface (CLI) provided by BD.

#### 2.3.3. Data Loading and FASTQ Processing

Quality control of the FASTQ files was performed using FastQC CWL 1.0 to ensure high-quality reads. The BD Rhapsody™ Sequence Analysis Pipeline was then used to process the sequencing data. This pipeline performed alignment of FASTQ reads to a mouse reference genome, which was provided as a pre-built tar.gz archive by BD. The analysis pipeline generated several outputs, including molecular counts per cell, read counts per cell, quality metrics, and an alignment file. The data were consolidated into a Seurat object containing a gene-expression matrix that linked each gene’s read count to its corresponding cell, along with annotations such as cell types and sample tags. This Seurat object was loaded directly into RStudio (version 2024.12.0.467) for downstream analyses.

#### 2.3.4. Gene-Expression Matrix Preprocessing and Analysis

The analysis was conducted using R version 4.4.1 and Seurat package version 5.1.0 [22], along with additional packages including dplyr, data.table, tidyverse, ggplot2, viridis, gridExtra, fgsea, msigdbr, RColorBrewer, colourpicker, and tibble. Several quality-control filters were applied to the initial Seurat object, which contained 18,102 cells. First, sample tags labeled as undetermined and multiplet were removed. Additional basic quality-control filters removed cells with fewer than 200 detected genes as potential empty droplets; cells with mitochondrial RNA content exceeding 50% were removed as damaged cells; and genes detected in fewer than three cells were removed as technical noise. Data normalization was performed using Seurat’s NormalizeData function, employing the global-scaling normalization method LogNormalize.

Genes that were highly variable across single cells were identified using Seurat’s FindVariableFeatures function with the “vst” method for variance stabilization. We selected 3000 highly variable genes to emphasize biologically meaningful signals in downstream analyses. Dimensionality reduction was performed using principal component analysis (PCA) via the RunPCA function. Based on the elbow-plot visualization of PCA standard deviations, the top 20 principal components were selected for subsequent analyses. Graph-based clustering was conducted using Seurat’s FindClusters function at high resolution to generate clusters containing 50 or more cells.

The clustered data were visualized using the two-dimensional Uniform Manifold Approximation and Projection (UMAP) algorithm implemented in Seurat’s RunUMAP function. Subsequently, we used the DoubletFinder package version 2.0.4 to identify and remove potential doublets [23]. Doublet rates for the BD Rhapsody system were based on manufacturer-provided cell-load numbers [24]. Approximately 3% of cells were identified as doublets and removed from further analysis. After quality-control steps had been applied, 12,443 high-quality individual cells were retained for further analysis. 

#### 2.3.5. Cell Type Annotation

Upregulated genes associated with each cluster were identified using a Wilcoxon rank-sum test implemented in Seurat’s FindAllMarkers function. Key cell-type-specific markers were examined to identify major pancreatic endocrine-cell populations. Beta cells were identified using *Ins1*, *Ins2*, and *Iapp* expression; alpha cells were identified by expression of *Gcg*, *Irx2*, *Arx*, and *Ttr*; gamma cells were identified by *Ppy* and *Arx* expression; and delta cells were identified by *Sst* and *Hhex* expression. Additionally, the Azimuth package version 0.5.0, a query–reference mapping algorithm for single-cell data, was utilized with the pancreas as the reference dataset to confirm cell-type annotations. The integration of both known molecular markers and reference-based annotation provided a robust identification of cell populations.

#### 2.3.6. Differential Gene Expression and GSEA

Analysis of differential gene expression between *Ide^+/+^* and *Ide^−/−^* samples was performed using the FindMarkers function with the DESeq2 test. The analysis parameters were set to identify genes that were differentially expressed between *Ide^−/−^* and *Ide^+/+^* populations using raw counts. This analysis was conducted on the complete dataset and separately on beta and alpha cells.

Gene set enrichment analysis (GSEA) was performed on the genes that were differentially expressed between *Ide^+/+^* and *Ide^−/−^* in both beta- and alpha-cell subsets using the clusterProfiler package version 4.12.0. The analysis was conducted using the clusterProfiler’s GSEA function with a *p*-value cutoff of 0.05 and the Benjamini–Hochberg method for multiple-testing correction. The Reactome subset of curated canonical pathways (C2 collections) from the msigdbr package was utilized [25]. Results were visualized using the enrichplot package version 1.24.0.

#### 2.3.7. Analysis of mRNA Splicing

Single-cell transcriptomic data were processed using a Velocyto pipeline (via command line, installed vi pip; version 0.17.17) to generate spliced, unspliced, and ambiguous RNA count matrices [26]. The spliced and unspliced matrices were used to compare gene-expression dynamics between the *Ide^+/+^* and *Ide^−/−^* conditions. Differential analysis was performed in Python (version 3.11.11) using the *t*-test from scipy.stats. Thresholds of 0.1 for log fold change and 0.01 for *p*-value were applied to determine significance.

## 3. Results

### 3.1. Proteomic Analysis of Islets of Langerhans from Ide^+/+^ and Ide^−/−^ NOD Mice

In a study designed to reveal the effect of Ide deficiency in the NOD model of autoimmune diabetes, we previously compared the bulk proteome of immune-infiltrated pancreatic islets from 10-week old (i.e., pre-diabetic) NOD mice. This analysis suggested a role of Ide in protein homeostasis, with evidence suggestive of an upregulated stress response in its absence; it also indicated an enrichment of pathways related to mRNA processing and splicing in Ide^+/+^ islet cells (see Figure 2J in [4]). A new analysis of the data identified enrichment of miscellaneous trafficking proteins involved in sorting between the ER and ER-associated degradation (ERAD) or related to Glut4 vesicles in *Ide^+/+^* islet cells (Figure 1A). In addition, a pathway analysis using Metascape software (version 3.5.20240901) provided strong suggestive evidence that proteins related to regulation of mRNA processing were far more (and almost exclusively) enriched in *Ide^+/+^* islets compared to proteins involved in any other pathway (Figure 1B), thus confirming our previous analysis carried out with an R interface to the Enrichr database (https://maayanlab.cloud/Enrichr/ accessed on 10 May 2025) [4].

Among the top 20 enriched proteins, proteins related to RNA processing included Zc3H11A, a protein involved in nuclear RNA export that associates with the TREX complex; Cstf3, which is required for mRNA polyadenylation; and Sap18 and Bud31, both of which are involved in pre-mRNA splicing (Figure 1A). Examination of a larger gene set related to RNA metabolism revealed enrichment of spliceosome components (Prpf38a, Srrm2, Cdc5l, Snrnp200), proteins involved in nuclear (pre-)mRNA export (Alyref, U2af2, Dhx9), biogenesis or import of small nuclear ribonucleoproteins (Prmt5, Tnpo1), and regulators of alternative splicing (Rbm25, Ptbp1) (Appendix A). Conversely, islet proteins from *Ide^−/−^* mice were enriched in ribosomal subunits, suggesting upregulation of translation consistent with islet hypertrophy in these mice [4].

### 3.2. Proteomic Analysis of Cytosolic IDE Interactants in HEK Cells

#### 3.2.1. Design and Verification of the Proximity Biotinylation System

The finding of a potential role of Ide in RNA processing was surprising and has not been reported before. Seeking to gain an initial mechanistic understanding of such a role, we reasoned that an analysis of the IDE interactome could reveal how the enzyme can affect mRNA processing. To explore the IDE interactome in the absence of potential confounding effects of the autoimmune inflammation in NOD pancreatic islets, we decided to analyze proteins that potentially interact with IDE in the cytosol of human embryonal kidney (HEK) cells using the TurboID proximity biotinylation approach [27]. We previously employed the same strategy to identify proteins that interact with an isoform of IDE targeted to mitochondria that is produced by translation from its first initiation codon (Met^1^) [20]; this analysis suggested interaction with mitochondrial ribosomes and respiratory-chain proteins.

Translation of IDE from its second in-frame initiation codon (Met^42^) has been reported to result in exclusive localization of proteins to the cytosol [28]. We fused a full-length human IDE cDNA starting at codon 42 to sequences encoding a V5 tag, the TurboId biotin ligase, a self-cleaving T2AP2A peptide, and tdTomato (Figure 2A top). The resulting construct was inserted into a lentiviral expression vector and used to transduce HEK293 cells that were then subjected to puromycin selection. Immunoblot analysis using an IDE antibody revealed bands corresponding to endogenous IDE (118 kDa) and to the fusion of IDE with the ligase (162 kDa), indicating correct and complete function of the self-cleaving peptide (Figure 2B, right panel). A parallel blot using a V5 antibody confirmed the presence of a major band at 162 kDa next to a minor band at about 140 kDa (Figure 2B, left panel). As this band was not stained with the IDE antibody, it likely corresponds to aberrant cleavage within the IDE protein N-terminal of the V5 peptide. To generate control cells, we transduced HEK cells with a lentivirus encoding V5, ligase, T2AP2A, and tdTomato but lacking IDE (Figure 2A bottom). V5 immunoblots revealed a single band corresponding to the expected molecular weight of the V5-tagged TurboId ligase (37 kDa) after removal of tdTomato by T2AP2A cleavage (Figure 2B, right-hand blot).

To verify that both constructs resulted in protein localization to the cytosol, we analyzed cells by confocal microscopy. This showed diffuse V5 staining over the entire cell, consistent with cytosolic localization of both proteins, with complete overlap of V5 and IDE staining for cells expressing the full construct (cyto-IDE) and exclusion of V5 staining from mitochondria visualized by staining for TOM20 in both Cyto-IDE cells and Cyto-TurboID control cells (Figure 2C).

Next, we examined the effect of biotin addition to cells using immunoblots stained with streptavidin–horseradish peroxidase (HRP) (Figure 3A). In the absence of added biotin, Cyto-IDE cells displayed staining of a background band at about 70 kDa; this band was enhanced upon biotin addition (Figure 3A, left panel). A band with the same approximate position also appeared in Cyto-Turbo-Id and untransfected HEK cells upon biotin addition (Figure 3A, center and right blots). Cyto-IDE cells also displayed bands corresponding to autobiotinylation of the IDE-TurboID fusion protein, presumably with biotin from the media. Addition of biotin to the media strongly enhanced autobiotinylation of the IDE-TurboID and of TurboID proteins and induced the appearance of multiple additional bands for both cell types. Affinity purification of biotinylated proteins from biotin-incubated cells using magnetic streptavidin beads resulted in strong enrichment of the 70 kDa background protein and the TurboID fusion proteins and the appearance of numerous additional bands (Figure 3A).

#### 3.2.2. Identification of IDE Proximity and Potential Interaction with CCR4-NOT

Having validated that exogenous biotin incubation resulted in enzymatic labeling of proteins, presumably including IDE interactants, we proceeded to affinity purification of biotinylated proteins from Cyto-IDE cells incubated with biotin using streptavidin-coated magnetic beads and analyzed purified proteins by mass spectrometry. To reduce background signals, we compared proteins purified from biotin-pulsed Cyto-IDE cells with two sets of control proteins: proteins purified from Cyto-IDE cells not pulsed with biotin and proteins from Cyto-TurboID cells pulsed with biotin.

Figure 3B shows a volcano plot of proteins purified from Cyto-IDE cells enriched relative to Cyto-TurboID cells, both pulsed with biotin. Intriguingly, proteins that were highly enriched in Cyto-IDE cells included three subunits of CCR4-NOT (CNOT2, CNOT3, CNOT8), a key complex expressed in all eukaryotes and present in the nucleus and the cytosol that regulates gene expression at all steps from the production of mRNAs in the nucleus to their degradation in the cytoplasm [29]. CNOT2 and CNOT3 are structural subunits of CCR4-NOT [18], although CNOT3 can also associate with ribosomes during stalled translation [30]. CNOT8 is involved in the control of mRNA levels through its deadenylase activity [19]. It is unclear why other subunits of CCR4-NOT, including the key scaffold unit CNOT1, which is expected to be a mandatory component of the complex, were not enriched. Although the CNOT proteins did not show the overall highest enrichment, CNOT8 and CNOT2 were enriched in Cyto-IDE cells relative to both controls and displayed the highest enrichment relative to the condition of Cyto-Ide cells without biotin (Figure 3C). Seeking to confirm the potential interaction of IDE with CCR4-NOT subunits through a second independent method, we used a proximity ligation assay (PLA) to probe the location of IDE relative to the three subunits. All three subunits showed highly significant proximity to cytosolic (and possibly nuclear) IDE (Figure 4A) at average frequencies of 27, 28, and 29 interactions per cell, respectively, for CNOT2, CNOT3, and CNOT8 (Figure 4B). Seeking to confirm direct Ide interaction with CCR4-NOT components, we also performed co-immunoprecipitation experiments that were not successful. This could be either due to inappropriate experimental conditions, which are frequently difficult to establish in co-precipitation experiments, or to the fact that Ide is close to CCR4-NOT but does not directly interact with it.

**Figure 3 cells-14-00792-f003:**
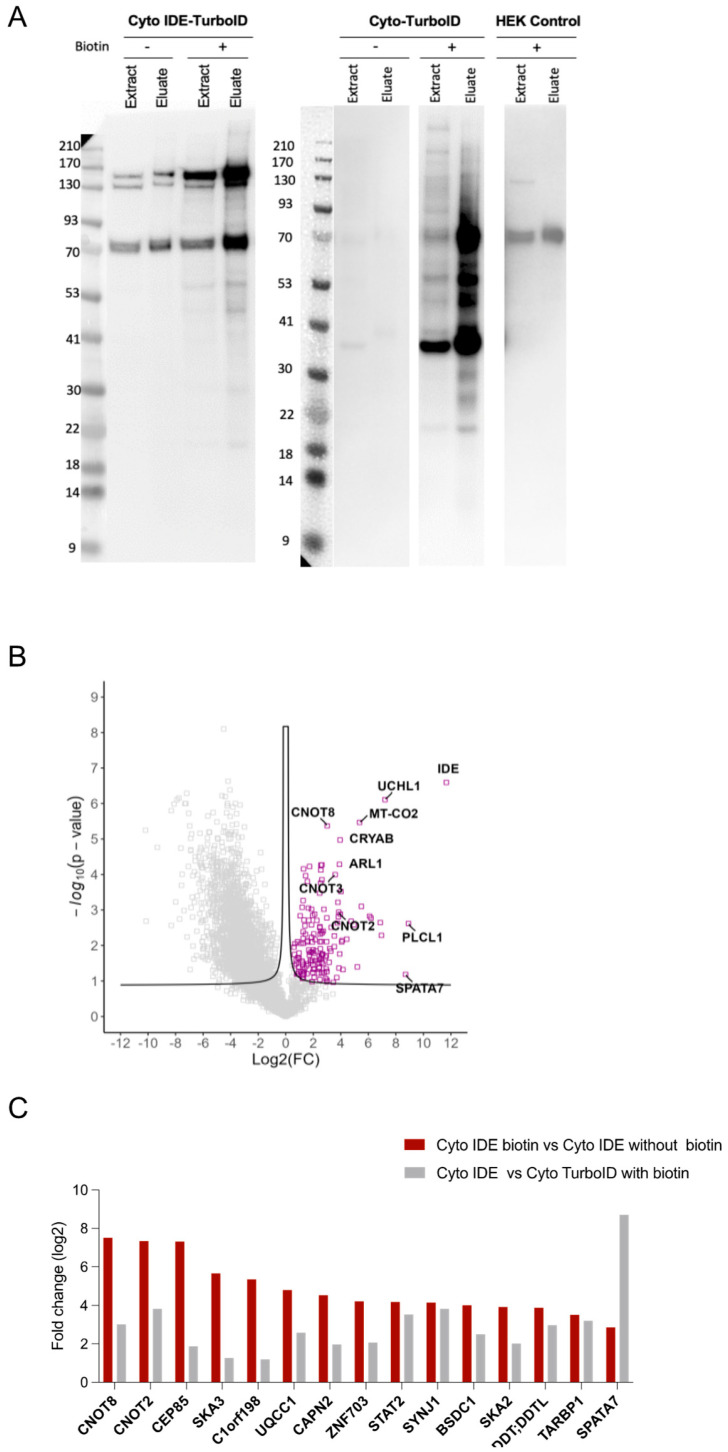
Identification of proteins interacting with IDE. (**A**) Immunoblot analysis of biotinylated proteins in Cyto-IDE (left), Cyto-TurboID, and untransduced HEK (right) cells incubated for 10 min with 50 μM biotin or buffer control, in total extracts and after enrichment using streptavidin beads. One out of four independent experiments is shown. (**B**) Scatterplot of biotinylated proteins enriched in Cyto-IDE vs. Cyto-TurboID cells incubated with 50 μM biotin. Data based on four independent replicates. Proteins with FDR=0.05 were analyzed. (**C**) 15 most-enriched proteins in Cyto-IDE cells relative to both controls (Cyto-IDE without biotin and Cyto-TurboId plus biotin).

### 3.3. Single-Cell Transcriptome Analysis of Ide^+/+^ and Ide^−/−^ Pancreatic Islet Cells

Our initial evidence suggesting a role of Ide in RNA processing was obtained through a proteomic analysis of *Ide^−/−^* and *Ide^+/+^* NOD mouse islet cells (Figure 1 and Appendix A). However, the islet proteome from female NOD mice is strongly imprinted by the presence of inflammatory lymphocytes and myeloid-cell infiltrate and by the stress in beta cells resulting from the metabolic challenge of insulin production in an inflamed environment. We wondered whether evidence linking Ide to RNA processing and translation could also be detected in islet cells from mice not undergoing autoimmune insulitis and the resulting metabolic stress. To address this question, we undertook a single-cell (sc) RNAseq analysis of islets from 12-week-old male *Ide^+/+^* and *Ide^−/−^* C57BL/6 mice fed standard chow. Islets were prepared from three mice per group and dispersed to single-cell suspensions that were pooled using equal cell numbers per mouse to a total of 3500 cells per genotype. The suspensions were processed for RNA extraction and transcription using Rhapsody ™ technology (Becton Dickinson).

scRNAseq data from 5923 sequenced cells were pooled and used to delineate endocrine and other (neuronal, stellate/ductal, and endothelial) populations through prior knowledge and data-driven gene sets. As expected, the endocrine populations were identified by expression of the key genes, as follows: *Gcg* (alpha cells), *Ins* (beta cells), *Sst* (delta), and *Ppy* (PP cells) [31]. Among beta cells, we identified a smaller population with high *Ins1* and *Ins2* (*Ins^++^* beta) expression and intermediate *Pdx1* and *Ide* expression. We also identified a major population (*Ins^+^* beta) with intermediate expression of *Ins1* and *Ins2* mRNA and high expression of *Pdx1* and *Ide* (Figure 5A and Appendix A). Heterogeneity among beta cells is a well-documented phenomenon and is thought to underlie, for example, the distinct responses of isolated beta cells to low glucose levels that trigger high insulin release only in a minority of beta cells [32]. A minor beta-cell population with uniquely high levels of insulin and proinsulin mRNA but low ratios of insulin protein to insulin mRNA (presumably due to upregulated degradation and/or secretion of the protein), termed extreme beta cells, was described by Farack and colleagues; these cells may be specialized for basal insulin secretion [33]. Analysis of differential RNA expression revealed that *Ins^++^* beta cells expressed higher levels of several cytosolic ribosome subunits (*Rpl12*, *Rpl35*, *Rps27l*) and of mitochondrial respiratory-chain proteins (*Uqcrb*, *Ndufa4*, *Ndufb1*) (Appendix A). Pathway analysis showed upregulation of translation initiation and elongation, as well as RNA processing and nonsense-mediated decay, in *Ins^++^* compared to *Ins^+^* beta cells (Appendix A).

Given that we had pooled equal numbers of *Ide^+/+^* and *Ide^−/−^* islet cells for scRNAseq analysis, we could evaluate the relative prevalence of each genotype in each pooled cell type. Interestingly, while *Ide* deficiency had no effect on the abundance of alpha and delta cells (50% per genotype), *Ide^−/−^* islets contained fewer beta cells but more Gamma/PP and immune cells than *Ide^+/+^* islets (Figure 5C). Nevertheless, the proportion of beta cells among all *Ide^−/−^* islet cells was only slightly reduced, despite a substantial increase in the number of gamma/PP cells among Ide^−/−^ islet cells (Figure 5B). Thus, although *Ide* deficiency induces islet hypertrophy and low-level beta-cell proliferation [4], it may reduce the number of beta cells slightly in favor of PP cells.

Our scRNAseq data also allowed us to assess *Ide* expression in different cell types. Although *Ide* was expressed in all populations, as expected, expression was increased in beta cells with intermediate levels of *Ins1/Ins2* mRNA (Figure 5D and Appendix A). Moreover, this population comprised a very limited number of cells with uniquely high *Ide* expression (Figure 5D), a feature consistent with confocal microscopy analysis of islets stained for Ide. A link between *Ide* expression and the global transcriptome of *Ins^+^* beta cells but not *Ins^++^* beta cells was evident in UMAP transcriptome projections of the two cell types, where *Ide* expression appeared to segregate only Ins+ beta cells in two populations (Appendix A).

To assess whether our observations of NOD islets and HEK cells could be reproduced in C57BL/6 islets, we performed a pathway analysis of genes enriched in *Ide^+/+^* as compared to *Ide^−/−^* islet cells. This was done separately for beta cells and alpha cells, the two most abundant populations in pancreatic islets. The transcriptome in *Ide^+/+^* beta cells was enriched for genes related to mRNA initiation and elongation, RNA processing, and nonsense-mediated decay (Figure 6A). Similarly, *Ide^+/+^* alpha cells were enriched for genes related to translation initiation and elongation and nonsense-mediated decay (Figure 6B).

Inspection of individual genes strongly enriched in *Ide^−/−^* or *Ide^+/+^* cells provided further insight into the potential biological roles of Ide. *Ide^−/−^* alpha cells were enriched for *Ppy* (encoding pancreatic polypeptide) (Figure 6D), suggesting that, in addition to increasing the proportion of Gamma/PP cells (Figure 5B), *Ide* deficiency might increase “promiscuous” expression of islet-cell hormones across subpopulations. *Ide^−/−^* alpha cells also contained more mRNA for *Scg5*, a chaperone for prohormone convertase, and *Ptprn*, a protein tyrosine phosphatase required for secretory-vesicle function and possibly indicative of alpha cell stress (Figure 6D).

Individual genes enriched in *Ide^+/+^* alpha cells included genes related to translation (*Eef1a1*, *Rps5*), protein degradation (*Ubc*), and proliferation (*Fos*, *Egr1*), the two latter genes also being enriched in beta cells (Figure 6C,D). Interestingly, *Egr1* and *Fos* mRNA was specifically enriched in all *Ide^+/+^* endocrine cells, with the exception of beta cells displaying high levels of *Ins* mRNA (Appendix A). Thus, in beta cells, *Ide* expression correlates with the expression of *Fos* and *Egr1*. Fos plays an important role in proliferation, signal transduction, and differentiation [34], and Egr1 is involved in cell survival and proliferation [35]. Although both genes are induced by signals stimulating growth, Egr1 can also suppress cell growth and transformation, and Fos expression is induced by the tumor suppressor p53 [36].

Finally, considering that our proteomic and transcriptomic data suggested that Ide might be linked to mRNA splicing, we analyzed our single-cell RNA data for evidence of altered splicing associated with *Ide* expression. This analysis identified a total of 111 gene transcripts that were differentially spliced/unspliced between *Ide^+/+^* and *Ide^−/−^* islet cells (Appendix A). While *Ide^+/+^* cells showed significant accumulation of spliced forms of some mRNAs and of unspliced forms of others (Appendix A left and right panels), the proportion of unspliced pre-mRNAs was slightly but significantly higher in *Ide^+/+^* cells (Appendix A). Interestingly, differentially spliced transcripts included several transcripts that were significantly enriched in *Ide^+/+^* (*Ide*, *Fosb*, *Ddx3y*, *Uty*, *Ped1c*) beta and alpha cells or in *Ide^−/−^* alpha cells (*CD59a*). The proportion of spliced *Ins1* (logFC = 0.153) and *Ins2* (logFC = 0.166) transcripts was higher in *Ide^+/+^* cells which however contained more unspliced *Ins2* transcripts (logFC = 0.247; Appendix A). Collectively, these results suggest that *Ide* expression may indeed affect splicing of selected transcripts, including *Ins1* and *Ins2*, although the details and mechanism of transcript selection remain unknown.

## 4. Discussion

Here, we describe evidence observed in both stressed and unstressed murine pancreatic islet cells and in a human tumor line at the levels of protein and mRNA expression, suggesting that Ide is involved in mRNA processing. Proximity biotinylation studies in HEK cells uncovered a potential lead regarding how Ide may affect mRNA processing, linking Ide to the CCR4-NOT complex. Consistent with a putative interaction between Ide and CCR4-NOT, *Ide^+/+^* islet cells displayed upregulation of pathways and proteins corresponding to the extensive array of CCR4-NOT functions, ranging from translation initiation and elongation to RNA nuclear export, mRNA decay, and protein quality control. However, the nature of the putative interactions between Ide and CCR4-NOT and their functional consequences remain for now unknown and subject of speculation.

Together with other observations related to Ide made in our laboratory and reported in the literature, the putative interaction with CCR4-NOT and the pathways affected by Ide expression observed in this study suggest a fundamental role of Ide in cellular mRNA and protein metabolism. By interacting with the ubiquitin–proteasome system, the principal mechanism mediating cytosolic protein turnover, and with CCR4-NOT, the principal cytosolic module for mRNA turnover, Ide could contribute to a regulation platform “controlling gene products from birth to death”, a role proposed for CCR4-NOT [37]. The possible Ide interaction with the CNOT8 subunit and the CNOT2/3 tandem supporting its function suggests that Ide might primarily be involved in a pathway degrading mRNA and proteins. However, since *Ide^−/−^* mice lack any striking phenotype in the absence of stress [38], any fundamental role of Ide in mRNA and protein homeostasis is likely limited to fine-tuning and to stress responses.

Our findings include some interesting evidence regarding a role of Ide in islet cells. Although *Ide* expression is clearly linked to pathways of mRNA processing, high *Ide* expression is found in beta cells with transcriptomes not strongly linked to RNA processing and intermediate *Ins* expression. It is tempting to speculate that Ide may be required to control mRNA processing and, through it, metabolism and insulin production in *Ins*^+^ beta cells, explaining that its loss results in beta-cell hypertrophy, unbridled mTOR activation, and (pro-)insulin production and diabetes in *Ide*^−/−^ C57BL/6 mice fed a high-fat diet [4]. Our observation that *Ide* expression modulates splicing of *Ins1* and *Ins2* transcripts suggests a potential mechanism contributing to this control.

Collectively, our findings suggest a model in which Ide may play a role in the third branch of the UPR, attenuation of protein translation. CCR4-NOT regulates protein translation and thereby expression levels through several mechanisms, including control of mRNA turnover and monitoring of mRNA stability and quality through direct links with ribosomes that involve CNOT3 [30,39]. We find that *Ide^−/−^* mice not only display enhanced mTOR-dependent proliferation of islet cells in the steady state, but also show extremely high insulin production and massive weight gain upon exposure to a high-fat diet [4]. Although an unhampered anabolic effect of high insulin levels might play a role in the latter phenomenon, we speculate that Ide might cooperate with CCR4-NOT to limit protein translation in stress situations, perhaps to an extent related to its relative expression level in each cell. Mechanistically, this could rely on the regulatory effects of an Ide-CCR4-NOT interaction or of cooperation between the deadenylase function of CCR4-NOT and proteolysis of small substrates by Ide, for example of protein fragments produced during stalled translation or nonsense-mediated decay; regulation of splicing by Ide-associated proteins could also be implicated.

Observations made by us indicate that Ide is induced in situations of proteotoxic stress and that its absence induces a moderate UPR both in pancreatic islet cells and in hepatocytes. Thus, Ide is required for coping with stress, a conclusion consistent with other literature reports [40,41,42]. How could Ide help cells to cope with proteotoxic stress? Given the limitation of its proteolytic activity to small substrates [43], Ide is not very likely to play a major role in upregulation of bulk protein degradation in an ERAD-like pathway, one of the three mechanisms cells use facing proteotoxic stress [44]. In contrast, in addition to cooperating with CCR4-NOT and/or splicing machineries, Ide may play a role in the second mechanism of the UPR, increased production of chaperones. Indeed, Ide has been proposed to act as a “dead-end chaperone” in stress-inducing conditions, although only a few substrates are known [9].

In conclusion, we provide evidence linking both murine and human Ide to control of mRNA processing and protein translation, potentially through a link with the CCR4-NOT complex and/or mRNA splicing. We propose a model of how Ide could control protein homeostasis and limit protein production in close cooperation with CCR4-NOT in settings of proteotoxic or metabolic stress. While this model is speculative, our findings suggest a novel and fundamental role of Ide consistent with its evolutionary conservation and will hopefully stimulate future research to elucidate the still-mysterious role of Ide in cell physiology.

## Figures and Tables

**Figure 1 cells-14-00792-f001:**
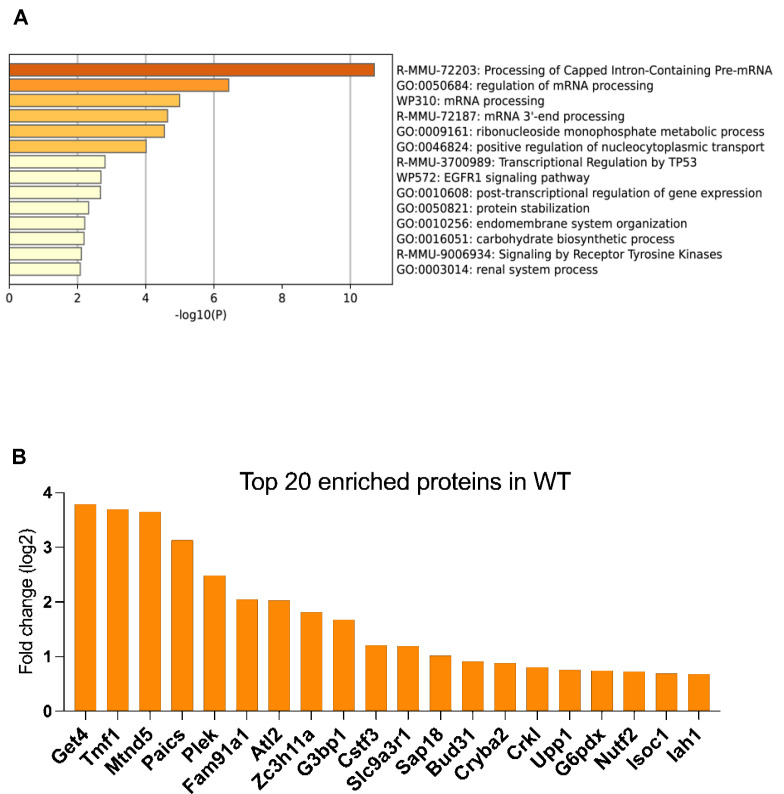
Proteomic analysis of islet proteins from *Ide^+/+^* and *Ide^−/−^* NOD mice. (**A**) Top nonredundant enrichment clusters, as identified by Metascape, among proteins upregulated in *Ide^+/+^* islets. The color scale represents statistical significance levels expressed as -log10. (**B**) Top 20 enriched proteins in *Ide^+/+^* islets.

**Figure 2 cells-14-00792-f002:**
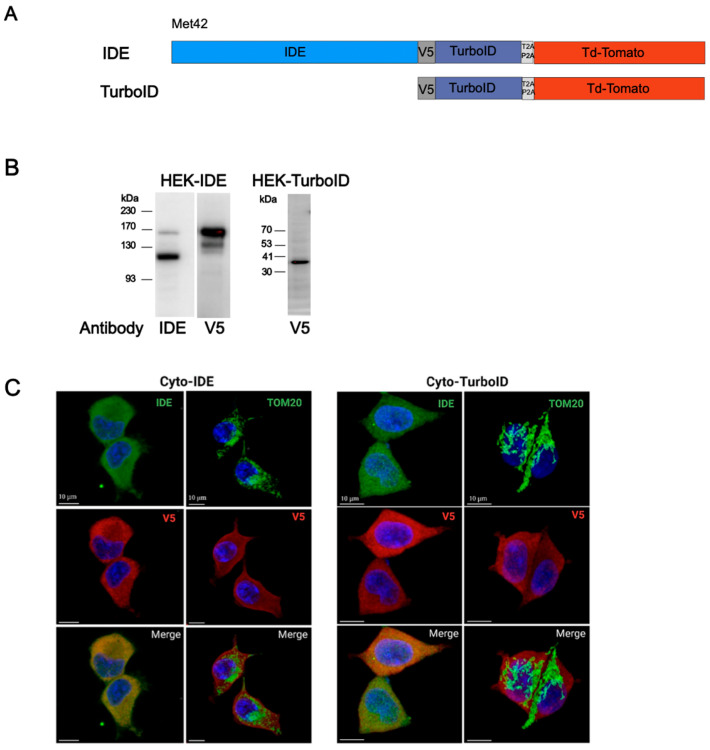
Characterization of TurboID fusion protein. (**A**) Schema of TurboID constructs for cytosolic expression. Human IDE cDNA starting at Met^42^ was fused to sequences encoding a V5 tag, the TurboID enzyme, the self-cleaving peptide T2AP2A, and tdTomato. (**B**) TurboID-fusion protein expression in HEK 293 cells was analyzed by Western blot in 10 μg total protein extracts using antibodies to IDE and V5 tag. (**C**) The expression of TurboID-fusion proteins in HEK cells was visualized by confocal fluorescent imaging. Cells expressing IDE–TurboID (**left panel**), or TurboID (**right panel**) were stained with antibodies to IDE, TOM20, and V5 tag. Merged images show cytosolic localization of each fusion protein.

**Figure 4 cells-14-00792-f004:**
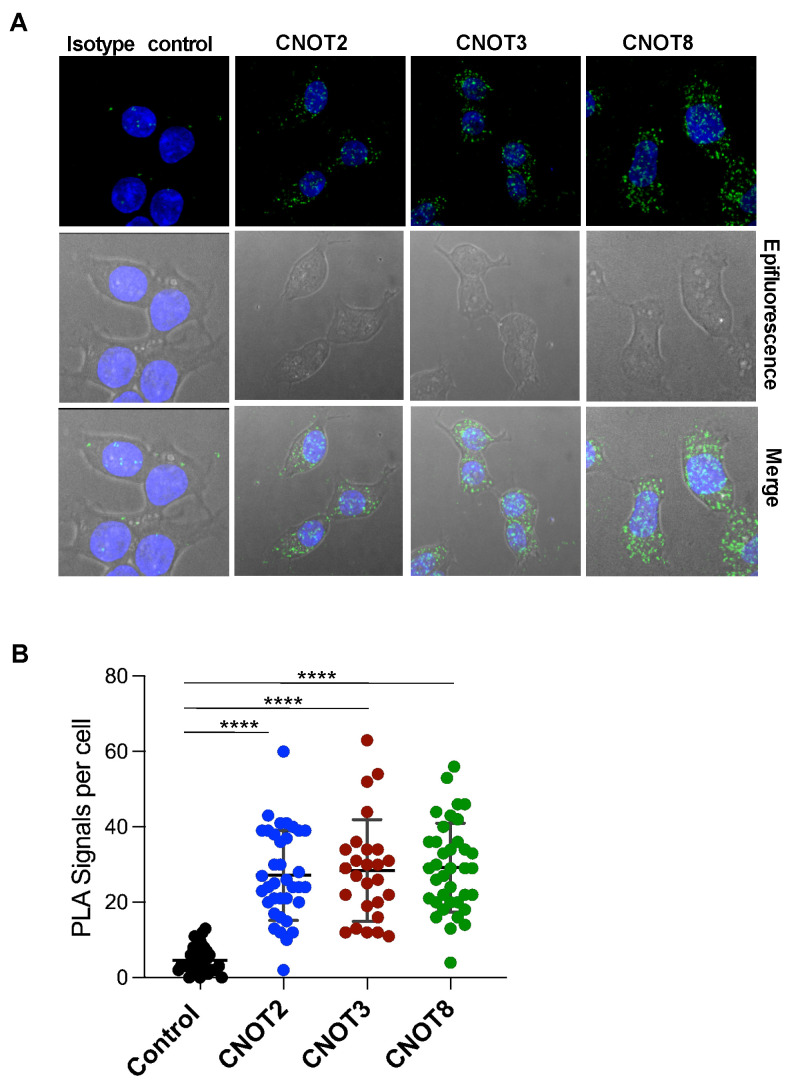
Ide may interact with the CCR4-NOT complex. (**A**) Proximity of IDE and CCR4-NOT was analyzed by PLA (green dots) using primary mouse antibodies to IDE and rabbit antibodies to CNOT2, CNOT3, and CNOT8. The cell area was outlined in epifluorescence images. Nuclei were counterstained with DAPI (blue). One of two independent experiments is shown. (**B**) Number of PLA positive signals per cell. The total numbers of cells analyzed were 36 (CNOT2), 26 (CNOT3), 37 (CNOT8), and 36 (isotype). ****, *p* < 0.001.

**Figure 5 cells-14-00792-f005:**
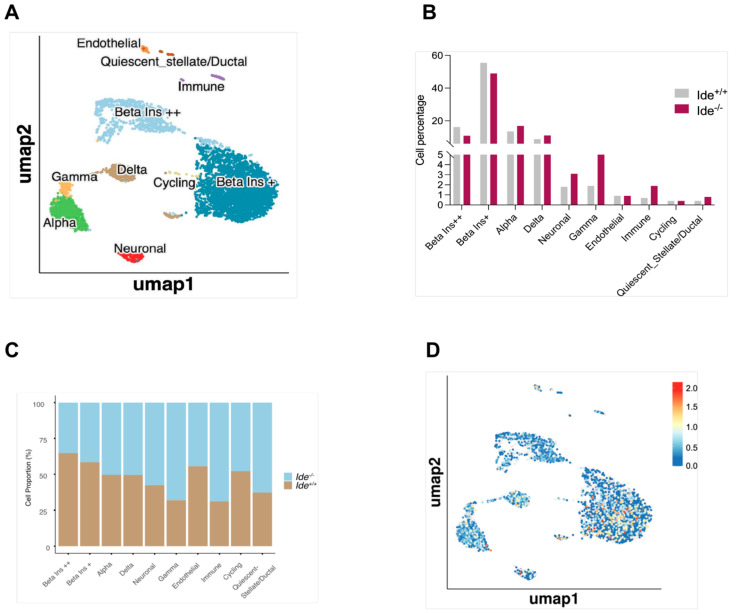
Single-cell RNA sequencing of *Ide^+/+^* and *Ide^−/−^* C57BL/6 pancreatic islets. (**A**) Uniform Manifold Approximation and Projection plot (UMAP) depicting clusters of cells from the two genotypes. Clusters are named and color-coded according to the legend shown at the right. (**B**) Percentage of each cell type among the total numbers of *Ide^+/+^* and *Ide^−/−^* cells. (**C**) Percentage of the two genotypes in each population identified in (**A**) among pooled sequenced islet cells. (**D**) *Ide* RNA expression levels in islet cell subpopulations.

**Figure 6 cells-14-00792-f006:**
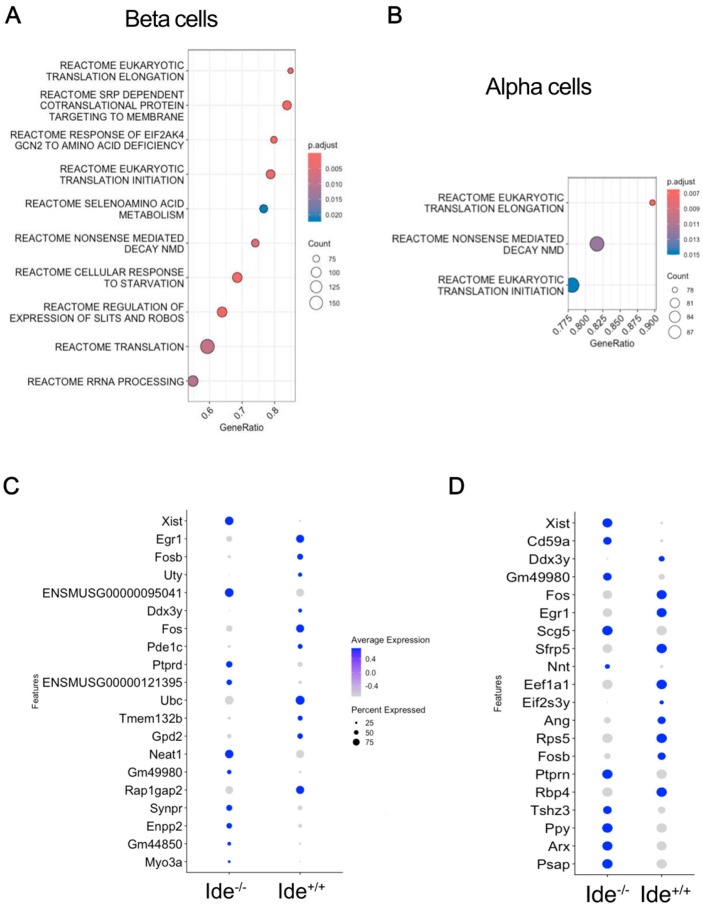
Single-cell RNA-seq analysis of beta and alpha cells indicates enrichment of genes related to translation and RNA processing in *Ide^+/+^* cells. (**A**,**B**). Pathways significantly associated with upregulated genes in Ide^+/+^ compared with *Ide^−/−^* cells (FDR < 0.05) were identified with Gene Set Enrichment Analysis (GSEA) with the Molecular Signatures Database (MSigDBr) Reactome gene sets. Gene Ratio refers to the proportion of upregulated genes among each gene set, and Count refers to the number of upregulated genes in the set. (**C**,**D**) Dot plots show the genes with the greatest expression changes in *Ide^+/+^* and *Ide^−/−^* beta (**C**) and alpha cells (**D**). Color indicates the average expression, and the dot size indicates the percentage of cells.

## Data Availability

The accession number for the proteomics data of NOD islets is XD034826 in PRIDE (EMBL-EBI); for the proteomics data of HEK293 cells PXD060228 in PRIDE (EMB-EBI); and for the scRNAseq data GSE299071 in GEO (NCBI).

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
