# Peer review of "Insulin-Degrading Enzyme Regulates mRNA Processing and May Interact with the CCR4-NOT Complex"

_cells, 2025, doi:10.3390/cells14110792_

Round 1
Reviewer 1 Report
Comments and Suggestions for Authors
Bertocci et al. propose an interesting study on an unexpected role of the insulin-degrading enzyme (IDE) associated with mRNA regulation that links the CCR4-NOT complex. They propose that IDE interacts with the CCR4-NOT complex, using the TurboID proximity biotinylation technique and by microscopy with proximity labelling assay. Using single cell RNA-seq, they showed an enrichment of genes associated to mRNA metabolism in IDE containing cells, thereby proposing a role for IDE in regulating mRNA stability that could be associated to the CCR4-NOT complex. Overall, the article is easy to follow, the experimental design and description of the figures are well done, and the conclusions are clear.
In the introduction, the authors have described in detail the role of IDE and the experiments performed in yeast, but I found the part about the CCR4-NOT complex a bit short (one sentence) and I think the manuscript would be more meaningful with a brief description of the implication of the complex in translation and homeostasis.
They identify an unreported interaction between IDE and 3 subunits of the CCR4-NOT complex, but they should discuss why the other subunits of the CCR4-NOT complex are not enriched in the TurboID proximity biotinylation assay and what this means for their model in the discussion.
I think it would be necessary for the authors to clarify the difference between the pathway analysis performed here and the analysis performed in Figure 2J (right panel) in reference 4.
For Figure 2B, it would be worth including a loading control so that protein expression can be compared between blots. At the bottom of the blot, mentioning "antibody" next to IDE and V5 might help the reader to better understand the figure. For Figure 3A, the images appear to be cropped and from different Western blots. It would be worthwhile to make this clear, for example by adding a space as in Figure 2B.
A general comment is that even in the numerical version, almost all the figures are at a very low resolution, for example we can hardly distinguish the named protein in Figure 3C.
Specific comments
For the title, I suggest removing “transcription”.
Line 26: instead of “protein expression”: “gene expression”.
Line 54: S. pombe in italic.
Lines 89-90: “the CCR4-NOT complex”.
Line 113: Verification instead of “Vérification”.
Line 304: (figure 2B, right panel).
Line 308: (figure 2A, 2B left panel).
Line 321: I think the “Volcano plot of Cyto-IDE proteomic data.” sentence should be removed as it is a scatterplot shown here.
Line 327: (Figure 3A, left panel).
Lines 327 to 329: I do not understand this sentence: “In the absence of added biotin, both untransduced HEK cells and Cyto-IDE cells displayed staining of a back- ground band at about 70 kDa that was enhanced upon addition of biotin.” as I only see biotin addition for HEK unstransduced cells.
Line 336: (Figure 3A, right panel).
Line 352: I do not quite understand why reference 27 was taken. I think that would be worth mentioning that CNOT3 can associate to the translating ribosome, if not mentioned in the introduction: doi: 10.1038/s41594-023-01075-8 .
Line 353: For the reference 28, it is not clear for me that the CNOT8 deadenyase activity is responsible of the mRNA upregulation in this article.
Line 359: (Figure 4A). Figure 4B should be described.
Line 365: PLA: proximity labelling assay
Line 420: Genes
Line 445: Erg1
Line 449: Erg1
Line 451: Erg1
Line 470: I would remove “nuclear” as the Ccr4-not complex is mostly described as important for cytosolic mRNA degradation.
Line 471: I am not sure that the term “recently” is appropriate as the study was published in 2014.
Line 489: I think the following reference is missing, doi: 10.1038/s41594-023-01075-8 .
Author Response
Please consult the document containing all replies

Reviewer 2 Report
Comments and Suggestions for Authors
The manuscript entitled Insulin-Degrading Enzyme Regulates mRNA Processing and 2 Interacts with the CCR4-NOT Transcription Complex 3 is interesting.BY combining proteomics and transcriptomic analysis together with biotynylation studies authors have shown the role of Ide interacting with CCR4-NOT and their role to cooperate to control 27 protein expression in proteotoxic and metabolic stress situations through cooperation between their deadenylase and protease functions. The study is well designed and techniques employed are sound.
Some of the concerns are
- If CCR4-NOT is silenced, does it impact on translational events?
- What is the effect of this interaction on mTOR signaling.
Author Response
Please read the document submitted in reply.

Reviewer 3 Report
Comments and Suggestions for Authors
In the present study, Bertocci B. et al. found a putative interaction between insulin-degrading enzyme (IDE) and the CCR4-NOT complex, a multi-protein complex that has polyA exonuclease activity and is involved in protein quality control. The authors also found an upregulation of pathways related to RNA processing and translation in Ide+/+ compared to Ide-/- islets. They also performed single cell transcriptome analysis of Ide+/+ and Ide-/- islets. The authors concluded that IDE regulates mRNA processing by interacting with the CCR4-NOT complex. The results are potentially interesting, but there are several questions that reduce the value of the present study.
Major criticisms
Proximity biotinylation analysis provides information about the molecular neighbourhood of a protein of interest and is not a method for detecting proteins interacting with the target protein. In particular, non-specific biotinylation has been reported with the TurboID enzyme used by the authors. Therefore, the authors need to confirm the interaction between IDE and CCR4-NOT proteins by other methods (e.g. conventional Co-IP experiments). In addition, CCR4-NOT is reported to have a strong deadenylase activity. What about the activity in Ide-/- cells?
Figure 1: Proteins involved in the regulation of mRNA processing were enriched in Ide+/+ islets. However, the authors did not directly test whether mRNA processing is affected in Ide-/- islets. Please show the expression levels of spliced (mature) mRNA and unspliced mRNA for some genes in Ide-/- islets.
Figure 5A: Two b-cell clusters (insulin++ and insulin+) were identified by single cell analysis. However, the existence of such 2 distinct clusters is unusual in control (C57BL/6) islets. Please show the differentially expressed genes between 2 clusters and perform the GO analysis.
Figure 5B: Please show the percentage of all cell types in each genotype.
Figure 6: Proteomic data suggest the changes of mRNA splicing in Ide-/- islets. Were the authors able to identify relevant changes in the single cell analysis?
Minor criticisms
Title: “interacts with the CCR4-NOT transcription complex” is not precise.
Author Response
Please read the document containing all replies

Reviewer 4 Report
Comments and Suggestions for Authors
Title: Insulin-Degrading Enzyme Regulates mRNA Processing and Interacts with the CCR4-NOT Transcription Complex
The main aim is to identify the potential role of insulin-degrading enzymes in cellular protein homeostasis through unbiased proteomics and transcriptomics approaches.
The draft was written excellently, and I must admire the manuscript's logical flow. Except for a few minor errors, the rest of the manuscript doesn’t need any further changes.
However, it is just a piece of advice and not to adhere strictly. As this article related to insulin and high-fat diet, I was wondering if the author could explore and depict the systemic effects of Ide+/+ and Ide-/- knockdown on mice's health as mice were terminated for 5 years. I was looking to see mitochondrial function and insulin resistance or mitochondrial dysfunction in mice after such a long period of living with Ide-/-. Please explain it and explore the role of a pathway that the author describes in how Ide could control protein homeostasis and limit protein production in close cooperation with CCR4-NOT in settings of proteotoxic or metabolic stress.
Best Luck!

Author Response
Please read the document containing our reply.

Round 2
Reviewer 3 Report
Comments and Suggestions for Authors
It is disappointing that the authors were unable to demonstrate the interaction between the IDE and CCR4-NOT components. However, they revised the manuscript properly according to my suggestions.